# Encryption Based Image Watermarking Algorithm in 2DWT-DCT Domains

**DOI:** 10.3390/s21165540

**Published:** 2021-08-17

**Authors:** Nayeem Hasan, Md Saiful Islam, Wenyu Chen, Muhammad Ashad Kabir, Saad Al-Ahmadi

**Affiliations:** 1School of Computer Science and Engineering, University of Electronic Science and Technology of China, Chengdu 611731, China; nayeemhasan@daffodilvarsity.edu.bd (N.H.); cwy@uestc.edu.cn (W.C.); 2Department of Software Engineering, Daffodil International University, Dhaka 1207, Bangladesh; 3Department of Computer Science, College of Computer and Information Sciences, King Saud University, Riyadh 11543, Saudi Arabia; salahmadi@ksu.edu.sa; 4School of Computing, Mathematics and Engineering, Charles Sturt University, Bathurst, NSW 2795, Australia; akabir@csu.edu.au

**Keywords:** blind image watermarking, digital information security, differential encryption, discrete wavelet transform, discrete cosine transform

## Abstract

This paper proposes an encryption-based image watermarking scheme using a combination of second-level discrete wavelet transform (2DWT) and discrete cosine transform (DCT) with an auto extraction feature. The 2DWT has been selected based on the analysis of the trade-off between imperceptibility of the watermark and embedding capacity at various levels of decomposition. DCT operation is applied to the selected area to gather the image coefficients into a single vector using a zig-zig operation. We have utilized the same random bit sequence as the watermark and seed for the embedding zone coefficient. The quality of the reconstructed image was measured according to bit correction rate, peak signal-to-noise ratio (*PSNR*), and similarity index. Experimental results demonstrated that the proposed scheme is highly robust under different types of image-processing attacks. Several image attacks, e.g., JPEG compression, filtering, noise addition, cropping, sharpening, and bit-plane removal, were examined on watermarked images, and the results of our proposed method outstripped existing methods, especially in terms of the bit correction ratio (100%), which is a measure of bit restoration. The results were also highly satisfactory in terms of the quality of the reconstructed image, which demonstrated high imperceptibility in terms of peak signal-to-noise ratio (*PSNR* ≥ 40 dB) and structural similarity (*SSIM* ≥ 0.9) under different image attacks.

## 1. Introduction

Sensors are used to capture different types of images for many practical applications such as clinical diagnosis, biometrics, and multimedia. The digital revolution has made it convenient to capture, store, and transmit images. However, with the rapid improvement of sensor technology, copyright violation, unapproved and illegal production, piracy, hacking and distribution, information theft, and several other statistical and differential attacks [1] have become important security concerns for the retrieval and distribution of digital images. The ever-increasing need for information security and intellectual property protection for digital images demands the development of robust watermarking techniques. Digital watermarking is the process of embedding information into a digital image (e.g., a digital image, video, etc.) to verify its ownership and authenticity by recovering the original information. There are many existing image watermarking techniques designed for various image security purposes. The trade-off between the robustness of the encrypted data and image quality is a significant challenge in digital watermarking research. The success of watermarking and de-watermarking processes depends on the successful retrieval of hidden data under different image processing attacks on the stego image [2,3]. Generally, the carrier image is not always enough to uphold the full length of the embedded data due to the degradation of image quality. Encrypting a watermark at a proper destination in an image can satisfy the main purpose of watermarking if it can endure different image processing attacks. A reliable digital watermarking technique should satisfy certain properties [4,5,6], e.g., imperceptibility of the watermark, robustness, security, and lower complexity with less error rate in the extraction phase.

Existing watermarking techniques can generally be divided into non-blind and blind techniques depending on the watermark extraction process. In the non-blind process, the original host image is required for watermark extraction, whereas, the blind watermarking process does not require the host image [7]. Non-blind watermarking techniques require more bandwidth for transmission, and sometimes, the carrier signal can be quite large for full extraction in partial transmission. Non-blind watermarking has limitations in real-time applications and is less effective relative to ensuring security [5,8] In contrast, the blind technique does not require the original signal for extraction, and it is more dynamic, convenient, and secure compared to the non-blind technique [9,10].

Blind watermarking can be performed in two different domains: (i) the spatial domain and (ii) the frequency domain. In the spatial domain, watermarks are embedded in the image pixels directly. Spatial domain methods are often simple but are less robust and fragile, and an image attack can easily destroy the watermark from the host image. In the frequency domain, the watermark is embedded into the transformed frequency channel, where the spatial pixel is not altered directly [9]. Frequency domain embedding techniques, (e.g., discrete wavelet transform (DWT) [11,12,13], discrete cosine transform (DCT) [14,15,16], singular value decomposition (SVD) [17,18], discrete Fourier transform (DFT) [10,19], and other decomposition methods [20,21]) are more robust and can utilize different characteristics of the signal to make the watermark more secure and imperceptible to the human eyes. The quality of a watermarking algorithm depends on the quality of imperceptibility and the recovery of the watermark signal. Generally, existing blind techniques often fail to extract 100% data on the retrieval phase from embedded data [22]. In most cases, the watermark data recovery is not cost-efficient due to the computational complexity of existing algorithms [23,24,25].

This paper proposes an encryption-based image watermarking scheme using a combination of second-level discrete wavelet transform (2DWT) and discrete cosine transform (DCT) with blind watermark extraction. Low-pass filtering attacks can hardly distort the image quality in that region. DCT transform is then applied to detect proper image pixels, which are less affected during extreme compression attacks. Unlike the existing method [22], we have utilized the same random bit sequence as the watermark and seed for the embedding zone coefficient. Since the seed is entangled for both the embedding location and the watermark sequence, full recovery of the watermark sequence is possible with authenticity in the extraction phase resulting in more secure watermarking. The proposed technique was tested under various types of attacks, and the results were verified with different measures, i.e., peak signal-to-noise ratio (*PSNR*), bit correction rate (*BCR*), and structural similarity index (*SSIM*). The main contributions of this work can be summarized as follows:In the proposed scheme, the 2DWT is selected based on the analysis of the trade-off between the imperceptibility and embedding capacity at various levels of decomposition. A large enough embedding capacity for the acceptable security of the watermarked image is also ensured.A technique is introduced to utilize the same random bit sequence as the cryptographic key and watermark for better security and convenience.The bit correction rate outperforms those of existing methods under different types of image-processing attacks on watermarked images.

The remainder of this paper is organized as follows: Related work is described in Section 2. The proposed scheme is presented in Section 3. The experimental and evaluation process is described in Section 4. Section 5 presents the experimental results and a corresponding discussion. Finally, the paper is concluded in Section 6.

## 2. Related Work

Here, we review state-of-the-art watermarking techniques with a focus on techniques in the transfer domain. We discuss the effectiveness and limitations of existing techniques to identify gaps in this crucial research area.

In the DCT-based watermarking scheme [8], multiple bits are used to embed watermarks in the central band of the host image by the corresponding coefficients of zig-zig operations. A sequential process from the left to right corners and from the top to bottom of the image is used for embedding. Roy et al. [8] revealed that this algorithm was comparatively robust under Joint Photographic Experts Group (JPEG) compression and noise addition. They proposed high embedding capacity watermarking, but it resulted in higher distortion in terms of image quality when a large number of bits were embedded in each block. The DCT values were computed for complete images in a block-based manner in Su at el. [26], which provided a low cost for image operations and was useful for the transformation of two-dimensional data representations. These criteria had achieved better solutions relative to image perceptivity and robustness; however, limitations of this method included relatively higher computational complexity and higher vulnerability of the embedded watermarks. Loan et al. [27] proposed a chaotic encryption-based image watermarking technique using the Arnold transformation, which was applied to differencing coefficient bits for the watermark operation. The outcomes of this algorithm demonstrated good robustness for different image operations, primarily related to compression, sharpening, cropping, and filtering. Ahmed et al. [28] also presented a watermarking scheme based on a nonlinear chaotic map with the orthogonal matrix. The scheme had advantages in regard to JPEG compression as well as noise tolerance.

The DWT approach is an effective process and is probably superior when watermarking is applied to the entire image [29,30]. Determining a favorable watermarking zone for bit embedding is a complex process, and it must proceed through various calculative operations for setting up. The combined DWT–DCT approach [31] provides generally more advantages on the image spectrum for watermarking. Yap et al. [32] proposed a new watermarking technique comprising a set of Krawtchouk moments on the image to select a local embedding portion. With this method, the produced watermarked image had been improved in robustness, especially for cropping attacks. Radeaf et al. [33] introduced an orthogonal polynomials-based watermarking technique that accumulates lower energy moments of the image using Krawtchouk–Tchebichef polynomials to hide the watermark data. This approach reduces the distortion in the host image and provides acceptable *PSNR* values. Qi et al. [34] proposed an adaptive region selection-based visible watermark on the host image. Although this method is a good solution for visible watermarking, for a higher texture image, it is often hard to find a salient enough region for watermarking to take place.

A bio-inspired watermarking procedure had been implemented in host images [35] with three-level DWT frequency transformation to increase the robustness criteria to some extent. Here, one-quarter of the watermark bits were implanted in the LL3 band, and the other bits were implanted in the remaining components of the transformed image. The good quality of the watermarked image was an advantage of this operation; however, the slow rate of embedding was a significant limitation for real-time applications, and the payload capacity was also comparatively low in the embedded image. Mohananthini et al. [36] proposed a genetic-based algorithm using the DWT and SVD domains for watermarking in a host image. Dong et al. [37] used logistics maps for encryption, and this scheme was designed for medical images. This watermarking scheme was tested under various attacks but resulted in poor payload capacity. A watermarking algorithm in [38] was introduced for monochrome images that enabled better security provisions for watermarked images; however, the robustness of this algorithm was insufficient against various types of image attacks. In addition, the computational cost increased due to the hybrid implementations of various encryptions.

Fares et al. [39] and Zhang et al. [40] proposed watermarking techniques based on the Fourier transform on various color channels. Multi-channel transformations created a comparatively higher risk of filtering attacks, and the colorimetric value changes over the image increased in complexity, too. Zhang et al. [40] proposed a blind watermarking technique that used a ghost imaging protocol. However, this encryption technique requires a complex environment, e.g., a computer-controlled digital micro-mirror device. This decomposition in higher frequency sub-bands (HH) is often significantly affected by JPEG compression, and the imperceptibility is reduced as the payload capacity increases for the embedding bits. Singh et al. [41] proposed a combined DWT–SVD method that divided an image into the two least and most significant bits for embedding, and this strategy was tested under some common image attacks, e.g., histogram equalization operations, and the *BCR* results were comparatively lower for the corrected bit rate. Fan et al. [42] proposed an algorithm based on the Gabor transformation and discrete cosine transform. This algorithm used Gabor transformation due to its scaling, direction, and optimization capabilities in image adjustment. This algorithm was robust against some image attacks, e.g., compression and filtering; however, the *PSNR* was not significantly improved. Lou et al. [43] presented a multi-scale watermark scheme based on Integer Wavelet Transform (IWT) and SVD. This approach achieved high imperceptibility quality, but the embedding process had a higher computational cost due to several extraction stages for the de-watermarking process. The resistance against several image attacks was satisfactory except in regard to the JPEG compression attack.

It has been observed that the majority of the existing methods are incapable of the satisfactory recovery of the watermark while ensuring minimal image disturbance. For example, Fares et al. [31] achieved 91% recovery for a salt and pepper attack, and Fan et al. [42] obtained 99.69% recovery for a Gaussian noise attack. When it comes to peer-to-peer communication or precise data retrieval, a single bit failure can cause total authentication failure. Hence, full recovery of the watermark is critical to ensure success, especially in an image attack environment. Thus, an algorithm proposing significant improvement of the recovery rate against common image attacks could be considered a remarkable achievement in the field of watermark research.

## 3. Proposed Scheme

The encryption-based watermarking technique embeds a sequence of random bits into suitable locations in the host image, and the recommended minimum length of the sequence is 128 bits [44] for reliable security. The two main issues related to watermarking are (i) the visual quality of watermarked image and (ii) its robustness against possible attacks. Although a longer bit sequence would increase the security, the increase in length would degrade the visual quality of the image. Different types of attacks on the watermarked image may also destroy the watermark, undermining the security proposed by the process. Common attacks include JPEG compression, low-pass filtering, noise, and geometric attacks.

To ensure the highest visual quality and robustness against possible attacks, the watermark should be embedded in a suitable location in the host image. Generally, the location is selected by the frequency domain transformation of the image [22,31], and the embedding capacity (i.e., the length of binary sequence) depends on the selected location, as discussed in Section 3.1. The watermarking algorithm is discussed in Section 3.2. The blind extract phase of the watermark is discussed in Section 3.3. Finally, the computing complexity is analyzed in Section 3.4.

### 3.1. Selection of the Distinct Embedding Region

One of the most serious concerns about the security provided by watermarks is that JPEG compression and other low-pass filtering attacks could destroy the watermark. All of these attacks suppress the high-frequency component of the image and intervene slightly on the low-frequency band [45]. JPEG compression especially increasingly suppresses the higher-frequency components with a higher degree of compression. Hence, to be robust against all of these attacks, the watermark could be embedded on the low-frequency components. However, embedding the watermark in this region could cause imperceptibility, which increases with the increase of the length of the random bit sequence.

DWT is a popular method to decompose an image to obtain the low-frequency components for watermark embedding [22,31]. DWT divides an image based on a basis function for different frequency spectra [46,47]. An image could be recursively decomposed into multiple levels. For a particular level (*j*), the image is first filtered with a low-pass filter (LPF) and a high-pass filter (HPF), and the filtered images are subsampled by a factor of two. The subsampled images are again filtered and subsampled to obtain LL*_j_*_+1_, LH*_j_*_+1_, HL*_j_*_+1_, and HH*_j_*_+1_ sub-bands, as shown in Figure 1. The low-frequency band (LL*_j_*_+1_) of an image is a very close approximation to the host image (I*_j_*), and it contains the basic frequency components of the host image. Several levels of DWT decomposition could be applied to the low-frequency band to obtain a relatively lower level frequency coefficient portion from the host image with the most significant energy compaction coefficients, which are less likely to be affected by low-pass filtering and geometric attacks.

With the increasing level of decomposition, the size of the LL band is reduced, which eventually decreases the embedding capacity. Hence, embedding a watermark to a particularly low-frequency band (LL*_j_*) introduces a trade-off between the imperceptibility and embedding capacity. While the increase in the length increases the security, at the same time, it decreases the perceptivity, which is measured as the peak signal-to-noise ratio (*PSNR*), as defined in Section 4.1. Figure 2 shows the average imperceptibility for different lengths of random bit sequences on different levels of the LL sub-band. This result was obtained by computing the average imperceptibility of several watermarked images (refer to Section 4) obtained by inserting the watermark into four different levels of the LL sub-band. The impact on imperceptibility for three different bit lengths was tested. It could be observed that LL1 and LL2 sub-bands pass the minimum threshold for imperceptibility (40 dB) and embedding capacity (128 bits). A compromise could be obtained by selecting a band that better preserves the energy of the signal to protect the watermark with the minimum thresholds for embedding capacity and imperceptibility.

In reality, the LL2 obtained by 2DWT provides better data preservation in low-pass filtering than that of the first level of DWT (i.e., 1DWT) [48]. The histogram (Figure 3) patterns of 1DWT and 2DWT reveal different effects in the low-pass filtering process. It should be noted that the magnitude fluctuation (y-axis) in 1DWT (500–5000 DWT coefficients) is greater than that of the 2DWT (300–1200 DWT coefficients). Thus, signal changes during the embedding process in 2DWT are comparatively small compared to 1DWT, which improves image quality. Additionally, the 1DWT signal yields less symmetrical and non-identical histograms compared to the host image histogram. In 2DWT, the embedding signal holds the symmetry to the host image histogram. The embedded zone in the lower frequency sub-band is less affected when a high level of compression is applied to the image [49], and the 2DWT watermark encoding minimizes distortion, edge effects, and discontinuity in the stego image.

To embed the watermark on the LL2 sub-band, the middle coefficient values are created (rather than changing the image coefficient directly), and discrete cosine transform (DCT) is applied to it. The DCT mainly calculates the weighted coefficient resulting from Fourier transform [50]. The DCT transform identifies the direct coefficient (DC) component of an image and the major definition of an image is originated from the DC component. As such, any changes in the DC coefficient cause a large distortion in the visual property. That is why the DC coefficient needs to remain unchanged during watermark embedding to maintain good visual quality. On other hand, the AC coefficients are the remaining coefficients with non-zero frequencies, and they carry lower magnitudes and represent spatial higher frequencies than those that have less of a visual effect in image quality. AC coefficients are less significant than DC in image reconstruction, and a small modification in these components can hardly be perceived by the human eye. In fact, a small perturbation in the AC coefficient is less likely to have any effect on the image properties. As such, the inclusion of watermark bits on AC coefficients in lower frequency sub-bands, on LL2 in particular, can achieve good imperceptibility quality and resist different attacks effectively. Hence, combining 2DWT–DCT transform yields possible optimal watermarking locations in the image by allowing minimal visual distortion and sufficient bit encryption, as shown in Figure 2.

### 3.2. Watermark Embedding Process

The host image for watermarking, denoted as I, is of the size *m* × *m*. The algorithm [22] embeds the watermarking bit into the lower sub-bands obtained by the 2DWT decomposition on the host image. Most of the image energy is concentrated at the lower frequency sub-bands [49]; thus, embedding watermark bits in that lower frequency sub-band is suitable for robust and steady watermarking. The embedding process is illustrated in Figure 4 and is described below.

Step 12DWT of the host image (I) could be obtained by two subsequent DWT decompositions. The first DWT is applied to I to create four different frequency sub-bands: LL1, LH1, HL1, and HH1. DWT is then applied again to the LL1 frequency coefficients matrix to obtain a lower frequency sub-band (LL2). The size of the LL2 sub-band is *m*/4 × *m*/4. The transformation is defined as follows:

LL1,LH1,HL1,HH1=DWT(I),

(1)LL2,LH2,HL2,HH2=DWT(LL1).

Step 2The lowest frequency sub-band (LL2) is selected to carry the watermark bits. Here, a vector is generated from the sub-band coefficient by using the zig-zag scanning operation [51]. The resultant vector coefficient is denoted **v***_n_*, where *n ≤ m/*4; *n × n* is the size of the LL2 sub-band.Step 3Vector **v*_n_*** is split into two parallel vectors **v_1_** and **v_2_** according to their positions on vector coefficients. Here, odd coefficients are stored in vector **v****_1_**_,_ and even coefficients are stored in vector **v_2_**:
(2)$$v_{1}\left(j\right)=v_{n}\left(k/2\right),\;v_{2}\left(j\right)=v_{n}\left(k/2+1\right),$$
where *k* = 1,..., *n*.

Step 4DCT is applied to each split vector (**v_1_**, **v_2_**) to produce two corresponding vectors **v**_1dct_ = DCT(**v****_1_**) and **v**_2dct_ = DCT(**v****_2_**), respectively.

Step 5A unified random bit sequence is created for watermarking and random position selection. The length of the sequence *W* = {1, −1}*^z^* is *z,* which is less than or equal to the size of the LL2 matrix (i.e., *z* ≤ *m*/4). To embed the *i*-th bit of the sequence, a random position *j*(1 ≤ *j* ≤ *m*/4) is selected using *W* as a seed. The watermark is embedded as follows:

(3)$$v_{1w}\left(j\right)=1/2\left[v_{1\mathrm{dct}}\left(j\right)+v_{2\mathrm{dct}}\left(j\right)\right]+\alpha W\left(i\right),\;v_{2w}\left(j\right)=1/2\left[v_{1\mathrm{dct}}\left(j\right)+v_{2\mathrm{dct}}\left(j\right)\right]-\alpha W\left(i\right)$$

Using Equation (3), we embed a bit in the same position of the two parallel vectors **v**_1dct_ (*j*) and **v**_2dct_ (*j*). The resultant vectors are denoted **v**_1w_ and **v**_2w_, respectively, and *α* is the watermark gain index for inserting the watermark. We avoid embedding in the first position of both DCT vectors to ensure that the DC property remains unchanged because changing the DC component generates significant distortion in the resulting image.

Step 6Inverse DCT is applied to vectors **v**_1w_ and **v**_2w_ to obtain **v′**_1w_ = DCT^−1^ (**v**_1w_) and **v′**_2w_ = DCT^−1^ (**v**_2w_), respectively. The two vectors are merged to a single vector, which is the inverse operation of Step 3 (i.e., embedding). Finally, the inverse zig-zig operation is applied to create a two-dimensional image matrix from a one-dimensional vector. This process is expressed as follows:

(4)$$\;\;v_{w}\left(k\right)={v^{\prime }}_{1w}\left(j\right),\;v_{w}\left(k+1\right)={v^{\prime }}_{2w}\left(j\right)\;\mathrm{for}\;k=1,\;\ldots ,\;n\;=1,\;\ldots ,\;n/2,$$

(5)$$\mathrm{LL}2_{w}\;=\;\mathrm{inverse}\;\mathrm{zigzig}\;\mathrm{of}v_{w}.$$

Step 7To reconstruct the final watermarked image, two of the inverse DWT operations are applied using LL2_w_ (in place of LL2) with other sub-band sets. The operations for the final watermarked image I*_w_* are expressed as follows:

(6)$$\mathrm{LL}2_{w}={\mathrm{DWT}}^{-1}\left(\mathrm{LL}2_{w},\;\mathrm{LH}2,\;\mathrm{HL}2,\;\mathrm{HH}2\right),\;I_{w}={\mathrm{DWT}}^{-1}\left(\mathrm{LL}1_{w},\;\mathrm{LH}1,\;\mathrm{HL}1,\;\mathrm{HH}1\right)$$

### 3.3. Extraction Process

The extraction process is illustrated in Figure 5 and is described below.

Steps 1–4These steps are the same as the embedding process (Steps 1–4). Here, sub-vectors **v**_1w_ and **v**_2w_ are obtained after completing Step 4.

Step 5The two sub-vectors are subtracted from each other, and the difference is denoted Δ**v**, where Δ**v** = **v****_1w_**(*j*) − **v****_2w_**(*j*) = *2W*. As seed *(W*) is known, we can calculate the random position (*j*) from the seed. Using a threshold on Δ**v**, we can then extract the watermark bits *W’* (Figure 6). Here, values less than 0 are extracted as −1, and the positive values are extracted as 1. Note that this extraction process is blind and does not require the original host image to recover the watermarked bits.

### 3.4. Computing Complexity

The algorithms for the embedding and extraction processes can be efficiently implemented in linear time. Table 1 shows the time complexity for different steps for the embedding and extraction of a key with length z in an image with the size *n* = *m* × *m*. Due to the use of a discrete convolutional kernel with a limited size, the DWT could be implemented in *O*(*n*) time. The watermark insertion/extraction process depends on the length of the key *k*. Generally, *k* is much smaller than *n*. All other steps take *O*(*n*) time.

Compared to the existing methods, the proposed method is simpler to implement. For example, the embedding process in [28] requires three steps (DCT, Gramm–Schmidt, and the nonlinear chaotic algorithm) in addition to a preprocessing step. For the nonlinear chaotic method, several levels of decomposition are performed that require additional computing time. Fares et al. [31] utilized DCT and 2DWT in addition to Schur decomposition during the encryption step and used three different bit insertion rules. The model proposed in [35] requires many steps, such as 2D permutation, 3DWT, the Just Noticeable Distortion (JND) mechanism, and the Genetic Algorithms. Sing et al. [41] used DWT–SVD and DCT algorithms together with the Arnold Cat Map.

## 4. Experiments

We have implemented the proposed algorithms in MATLAB, and the source code is available on GitHub (https://github.com/NayeemHasanT/Digital-Watermark, accessed on 3 August 2021). The proposed watermark embedding process was evaluated with different types of images. Here, the cover images used for testing were collected from the USC-SIPI Image Database (http://sipi.usc.edu/database/, accessed on 3 August 2021). Sample images of different sizes were used in these experiments. For example, the baboon, bridge, jet-plane, boat, sailboat, and pepper images are 512 × 512 pixels, the girl image is 256 × 256 pixels, and the pirate image is 1024 × 1024 pixels. The watermark used for encryption was a random sequence of 256 bits, a typical payload [24], which was produced using a pseudorandom generator. The MATLAB programs for the watermark embedding and extraction processes were executed in a Windows 10 (64-bit) environment by using a personal computer with AMD (Ryzen 3 3200G) 3.6 GHz processor and 5.95 GB of RAM. Table 2 shows the execution time for images of different sizes.

### 4.1. Performance Metrics

The peak signal-to-noise ratio (*PSNR*), bit correction ratio (*BCR*), and structural similarity metrics (*SSIM*) were used to evaluate the imperceptibility of the watermark and the robustness and quality of the watermarked image. These measurements are common metrics used in watermarking [52].

*PSNR* is often used as a quality measure between original and modified images [23], which is defined as follows:(7)$$PSNR=10\mathrm{lo}g_{10}\frac{Max^{2}}{MSE}$$
where *Max* is the maximum pixel value in the original image, and *MSE* represents the error between two *m* × *m* sized images (i.e., the original and watermarked images), which is defined as follows:(8)$$MSE=\frac{1}{m^{2}}\overset{m}{\underset{i=1}{\sum }}\overset{m}{\underset{j=1}{\sum }}{\left[I_{1}\left(i,j\right)-I_{2}\left(i,j\right)\right]}^{2}.$$

For de-watermarking, a *PSNR* value greater than 40 dB is an indicator of good quality image reconstruction [52].

The *SSIM* is a perceptual metric (defined by Equation (9) that quantifies image quality degradation caused by processing, e.g., data reconstruction or compression. *SSIM* measures the perceptual difference between two similar images and gives a quality reference by comparing the original and modified images.
(9)$$SSIM=\frac{\left(2{µ}_{1}{µ}_{2}+C_{1}\right)(2\sigma \left(I_{1},I_{2}\right)+C_{2}}{\left(µ1^{2}+µ2^{2}+C_{1}\right)\left(\sigma {\left(I_{1}\right)}^{2}+\sigma {\left(I_{2}\right)}^{2}+C_{2}\right)}.$$

Here, C_1_ and C_2_ are constants that ensure stability when the denominator becomes 0, *µ*_1_, *µ*_2_ is the mean value, and *σ* is the variance value of images I_1_ and I_2_.

The *BCR* measures the accuracy of the extracted bits (Equation (10)). The *BCR* compares two binary sequences, i.e., the inserted (W) and extracted (W′) watermarks. *BCR* is the ratio of the correctly extracted bits over the total number of embedded bits in percentage.
(10)$$BCR=\frac{1}{z}\overset{z-1}{\underset{k=0}{\sum }}\left(\bar{W\left(k\right)⊕W(k}\right))\times 100.$$

Here, *z* is the length of the bit sequence and ⊕ is the XOR operator. The *BCR* value is 100% if the watermark is extracted without any bit error.

### 4.2. Effect of Gain Factor

There is a trade-off between invisibility and robustness in image watermarking. To maintain a good balance between these qualities, a suitable gain index (*α*) value should be selected for embedding. We experimented with three gain index values (*α* = 0.1, 0.2, and 0.3), and the results are shown in Table 3. As it can be seen, for *α* = 0.1 and *α* = 0.2, the *PSNR* values are greater than the acceptable margin for image imperceptibility (i.e., 40 dB) for all images, which was not obtained for the value of *α* = 0.3. The *PSNR* values are comparatively lower for higher gain index values; however, the *BCR* of the restored watermark remains intact. Thus, to maintain visual quality and robustness in watermarking, a balanced value of *α* = 0.1 and *α* = 0.2 was used in all subsequent experiments.

### 4.3. Embedding Capacity

As the length of the random bit sequence *W* (used as the watermark) increases, the security of the encryption increases; however, the image quality degrades. We examined the embedding capacity (number of bits) of the proposed watermarking scheme to validate its effectiveness. Figure 7 shows the *PSNR* values for different bit lengths. We found that the benchmark value of 40 dB could be achieved for all images with a bit stream size of 256–512 bits and near 40 dB for 1024 for some images.

## 5. Results and Discussions

We evaluated the proposed method on different images for robustness, imperceptibility, and *BCR* against various types of image attacks (e.g., compression, filtering, and geometrical, cropping, and histogram attacks). We tested different watermarked images, as shown in Figure 8, along with their resultant *PSNR* values using gain factor *α* = 0.1 and a bit stream length of 256 bits. As shown in Table 3, the *PSNR* values of the tested images were greater than 40 dB, and the *BCR* was 100% for the tested images, which is a significant outcome. In addition, the *SSIM* values were greater than 99%, except for the girl image, which had the minimum resolution (256 × 256).

### 5.1. Resistance to JPEG Compression Attacks

Table 4 shows the *BCR* (%) values of the recovered bits under different JPEG compression attacks for different watermarked images. Here, the quality factor (Q) is the JPEG compression quality strength, which varies from 10 to 70 [25]. It could be observed that our watermarks resist deep compression attacks. For different values of the quality factor Q ≥ 30, the watermark bits were recovered completely (100%) under highly compressed JPEG attacks for all the images, except the pepper image. The pepper image has a texture difference that is more affected by JPEG compression [40,53]. The proposed scheme uses a higher frequency level (2DWT) for embedding, which makes it more robust against JPEG attacks than existing methods [22] that only use single-level transformation.

### 5.2. Robustness against Common Noise Attacks

Different common attacks, e.g., Gaussian, salt and pepper, and speckle noise addition attacks were examined (Figure 9), and we achieved successful watermark extraction (Table 5) while preserving high *SSIM* values (Figure 10). Gaussian, salt, pepper, and speckle noise attacks primarily affect a particular region in the spatial data. As the proposed method employs frequency domain embedding, the watermarks are protected under two transform layers, and encrypted bits are barely affected in the lower sub-band zone. Under such high-variance noise attacks, we obtained nearly 100% *BCR* for most of the tested images.

It can be observed from Figure 10 that the *SSIM* is relatively lower for the jet plane and peppers images. These two images are with a single-color priority that is widely spread across the photos. For example, the color white is dominant in the jet plane image, whereas green and red are dominant in the pepper images. Any changes to the matrix coefficient represent high-effect changes in variety in the *SSIM* equation relative to the composite image of different colors, such as in the baboon, pirate, and bridge images.

### 5.3. Robustness against Image Enhancement Processes

We examined different types of image enhancement processes and bit removal activity relative to *BCR* for the reconstructed images. Frequency enhancement operations primarily suppress the pixel value by shortening or enhancing image intensity. It should be noted that the edge and smoothing areas of the image are primarily affected by these operations. The proposed method employs the zig-zig scan, which characterizes edge coefficients in the tail portion of the vectors; thus, the filtering operation cannot change the watermarked data in the embedding region. The bit removal attack in the less significant bit does not change the major difference of the original pixel value significantly. Instead of a direct pixel coefficient, we inserted a watermark into middle coefficients obtained from the parallel vectors (Equation (3)). Table 6 shows the *BCR* results after enhancement attacks (bit removal, gamma correction, and sharpening). Extraction under the bit removal process achieved satisfying *BCR* results (achieving 100% for most of the test) compared to the existing methods [40], which validates the proposed scheme’s suitability for such types of intervening.

### 5.4. Robustness against Cropping and Geometrical Attacks

Cropping and geometrical transformation are common attacks in scan and print processes. The proposed method demonstrates resistance against bit plane removal, cropping, and geometrical attacks (Figure 11). Although the proposed scheme is limited against a greater rotational effect, it is very robust against rotational mechanisms. In rotational attacks, the image pixels are translated at an angle difference, and symmetric resizing (512 to 256 to 512) returns the original pixel value. The cropping process cannot easily omit the watermark because DCT spreads the watermark all over the image rather than to a particular region. Thus, data loss is lower in rotating and cropping operations. The symmetric resizing process (512 to 256 to 512) is fully robust with no bit error; however, asymmetric resizing (512 to 200 to 512) is less robust after regeneration processes. Table 7 shows the *BCR* for rotational, resizing, and cropping modifications.

### 5.5. Robustness against Different Filtering Operations

Different image filtering attacks are imposed into our watermark image, and the *SSIM* and *BCR* values obtained by the proposed method were examined (Table 8). Typically, filtering operations cause a linear modification on the image pixel. The embedding zones in the proposed method are split into two parallel vectors, and the linear change due to filtering is concurrently affected on two parallel vectors. As such, the subtraction of the corresponding vector coefficients (extraction phase) gives the same value before and after filtering attacks. This means that the attack is prevented by the proposed mechanism successfully, and the watermark is extracted from filtering attacks. The *BCR* values were 100% for most of the tested images under various filtering operations.

Additionally, we have tested the proposed scheme on different levels of DWT composition for different filtering attacks (Table 9) to justify our selection of a 2DWT sub-band. Figure 12 shows the differences between the existing 1DWT [22] method and the proposed 2DWT watermarking method for different image attacks. The proposed method achieved better results in most cases using 2DWT.

### 5.6. Comparison with Other Methods

The main concern of an encryption-based watermarking technique is the full extraction of the watermark by ensuring the optimal quality of the watermarked image. Hence, the proposed method outperforms the existing methods in terms of *BCR*, as shown in Table 10. The proposed method obtained better results (*BCR*) than the methods proposed by Ferda et al. [54], Feng et al. [55], and Jiang et al. [56] under salt and pepper and JPEG compression (quality factors of 20 to 60) attacks. With an extreme JPEG compression ratio (quality factor 20), the proposed method outperformed the existing methods. It could be noted that Lin’s scheme [57] obtained similar results; however, this scheme is non-blind, and the original host image is required to extract the watermark. The average *PSNR* (dB), values (Table 11) obtained by our method are higher than the minimum threshold (40 dB) and the method also outperformed the existing methods. The higher *BCR* and *PSNR* value obtained by the proposed method demonstrates its practical effectiveness and suitability for protecting image properties while preventing attacks.

It is observed that among the compared methods, Feng et al. [55] achieved higher *SSIM* than the proposed method, but it failed to obtain a good enough *BCR* value (e.g., 69.80 for JPEG compression) for credible image verification. In addition, Lin’s scheme [57] applies watermarking techniques with quantizing difference map values, and watermark bits are embedded into the block-based wavelet coefficient. Compared to Lin’s scheme, the proposed scheme achieved a higher *BCR* and can minimize computational complexity.

## 6. Conclusions

In this paper, a robust and secure watermarking scheme based on the encryption of a random binary sequence is discussed. The proposed method uses 2DWT–DCT, a combination of two frequency domain techniques, where the second level of wavelet transform enables higher protection capabilities and ensures large enough embedding capacity for acceptable security at the same time. The proposed scheme was tested under various image processing attacks. The experimental results indicate that the proposed method performs well under different types of image processing operations, e.g., image filtering, compression, sharpening, bit removal, and noise addition attacks. In addition, we have used a parallel vector, rather than using a single vector, to minimize the *PSNR*.

Attackers often try to impair watermarks by estimating susceptible watermark locations from coefficient correlation. To minimize the risk of this type of attack, the length of the watermark sequence can be increased as a future work. The error correction code can also be added to multiple frequency levels with watermark bits to maximize bit recovery in interrupted image transmission. This algorithm is also extendable for the watermarking of color images. If a color image is decomposed into three channels in RGB format, we can obtain a larger space for the watermarking, yielding the opportunity to embed a longer cryptographic key and achieving better imperceptibility as well. Furthermore, a biometric-based cryptography key [66,67] could be considered for the enhancement of watermark security. All of these could be considered as future works.

The main advantage of the proposed scheme is that the bit rate error of extracted watermark is extremely low compared to recently proposed watermarking schemes. The watermark cannot be easily omitted by tampering with any area of an image. In addition, the imperceptibility of the proposed scheme was satisfactory. Thus, we conclude that the proposed scheme is well suited for copyright protection, ownership verification, and different cybersecurity applications. The proposed scheme can be used to protect the integrity of medical images and to preserve biometric data. Finally, the proposed method can be incorporated in the digital signature, photography identification, and other internet security tasks.

## Figures and Tables

**Figure 1 sensors-21-05540-f001:**
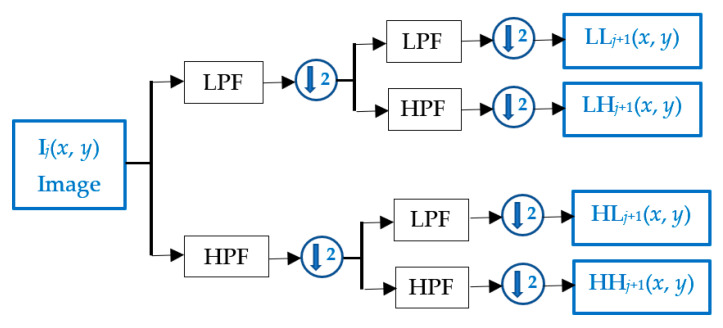
DWT of an image I from level *j* to *j* + 1 producing sub-bands LL*_j_*_+1_, LH*_j_*_+1_, HL*_j_*_+1_, and HH*_j_*_+1_.

**Figure 2 sensors-21-05540-f002:**
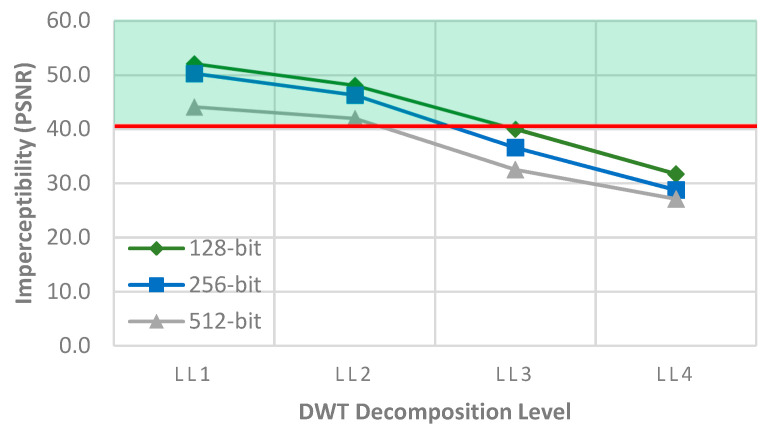
Trade-off between imperceptibility and embedding capacity in various LL sub-bands.

**Figure 3 sensors-21-05540-f003:**
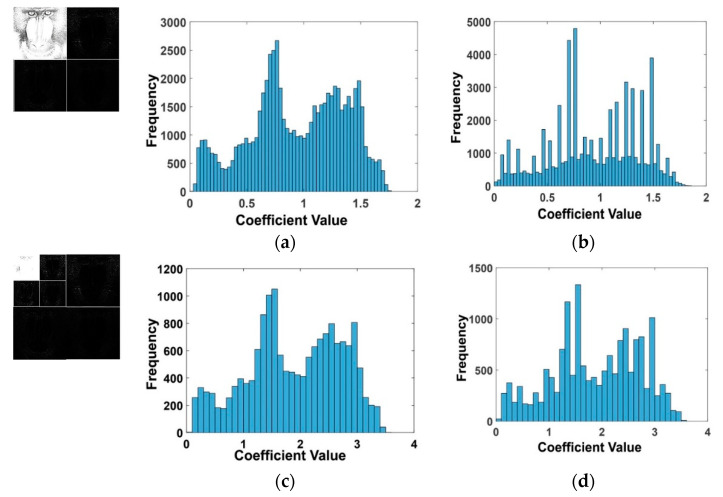
Histogram analysis before and after embedding baboon image in 1DWT and 2DWT: (**a**) 1DWT histogram before embedding, (**b**) 1DWT after embedding, (**c**) 2DWT histogram before embedding, and (**d**) 2DWT histogram after embedding.

**Figure 4 sensors-21-05540-f004:**
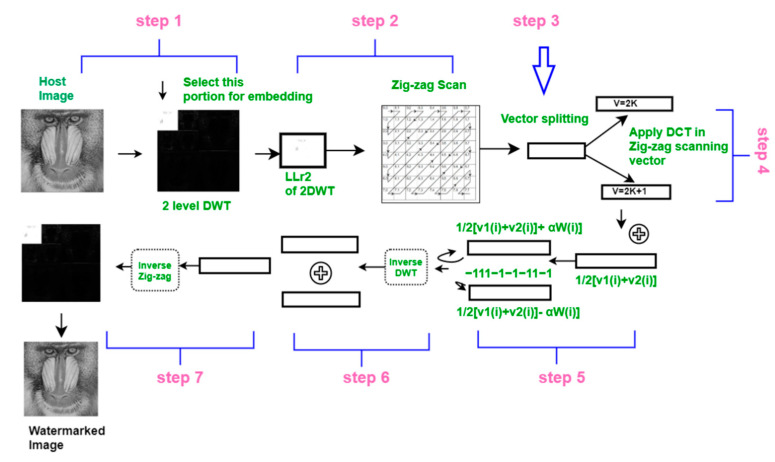
Embedding procedure of 2DWT-DCT watermarking.

**Figure 5 sensors-21-05540-f005:**
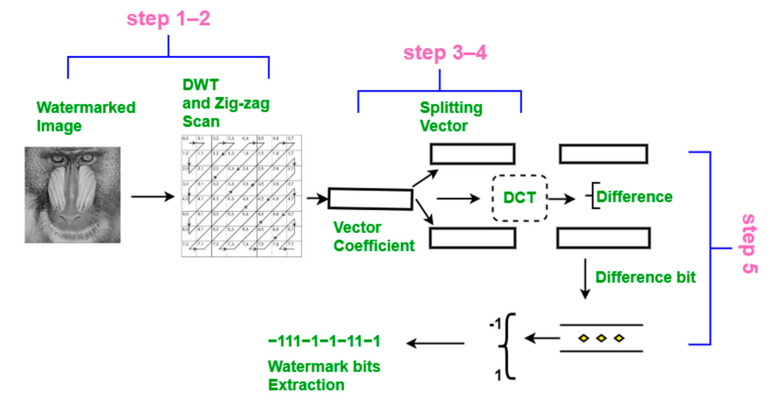
Extraction procedure.

**Figure 6 sensors-21-05540-f006:**
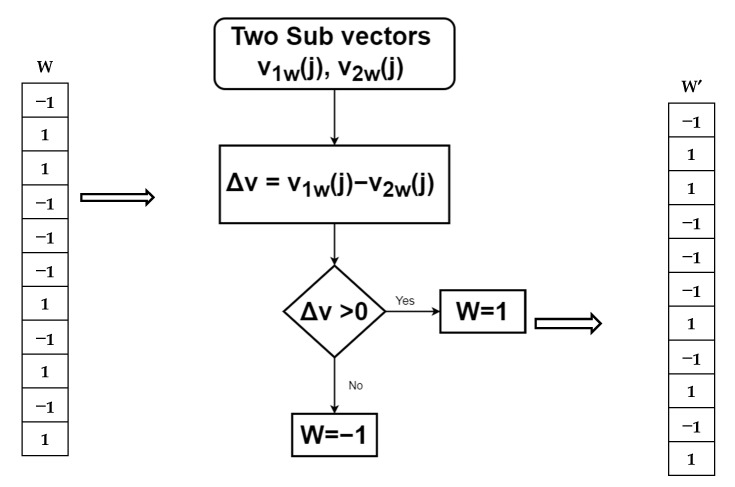
Bit extraction from subtracted values of splitting vectors.

**Figure 7 sensors-21-05540-f007:**
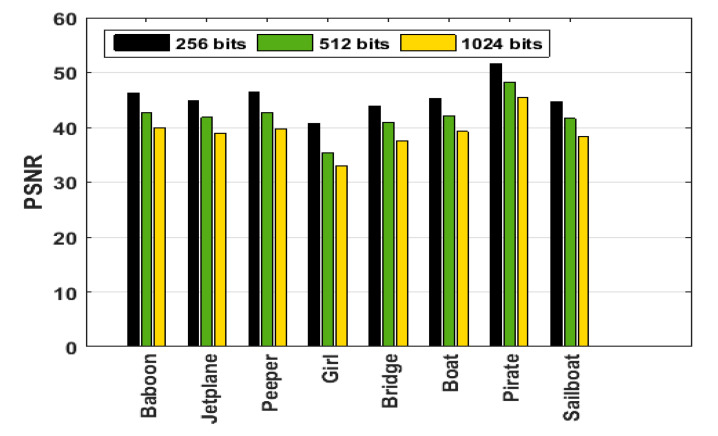
*PSNR* (dB) with different bit lengths as random bit sequences using *α* = 0.1.

**Figure 8 sensors-21-05540-f008:**
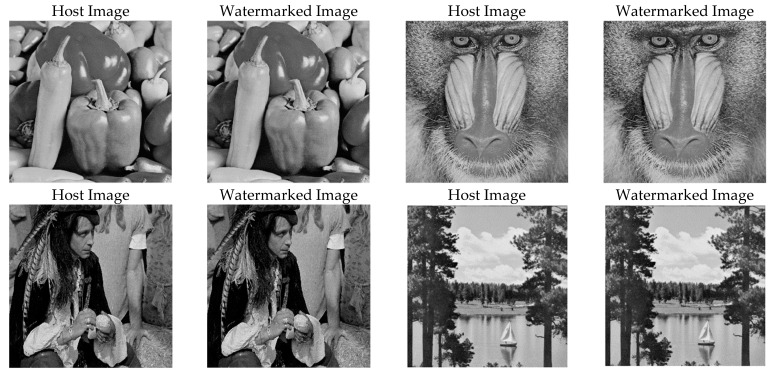
Original images and corresponding watermarked images. The *PSNR* (dB) values of the resulting watermarked images are 46.8532, 45.6986, 52.7633, 46.7466, for the pepper, baboon, pirate and sailboat, respectively.

**Figure 9 sensors-21-05540-f009:**
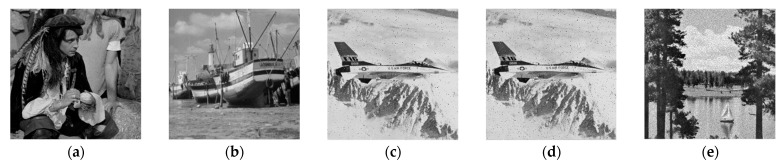
Watermarked images with common noise attacks (**a**) Gaussian filter (5 × 5) var = 1.5, (**b**) average filter (3 × 3), (**c**) salt and pepper noise (var = 0.01), (**d**) salt and pepper noise (var = 0.02), (**e**) speckle noise (var = 0.02).

**Figure 10 sensors-21-05540-f010:**
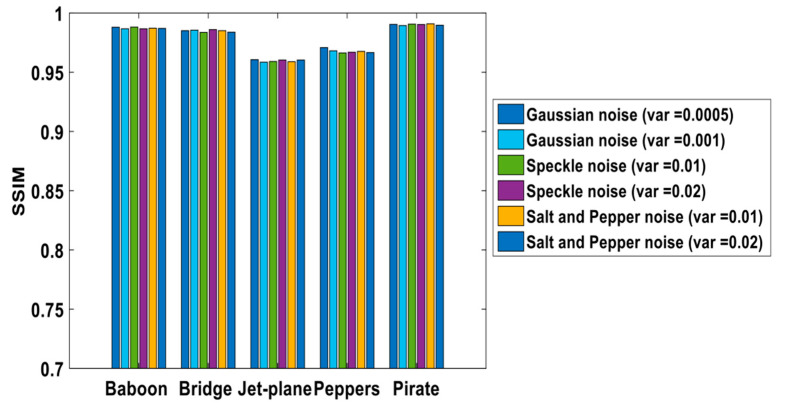
*SSIM* Value for Different noise attacks.

**Figure 11 sensors-21-05540-f011:**
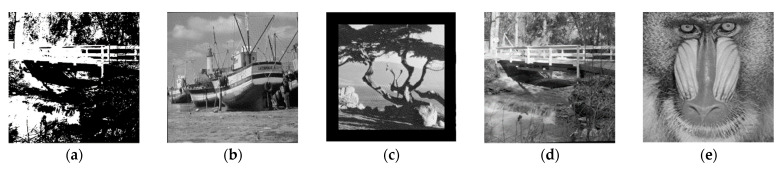
Watermarked images with geometrical attacks (**a**) bit plane removal, (**b**) rotation 0.25 degree, (**c**) surrounding crop 15%, (**d**) resizing 512 to 256, and (**e**) resizing 512 to 200.

**Figure 12 sensors-21-05540-f012:**
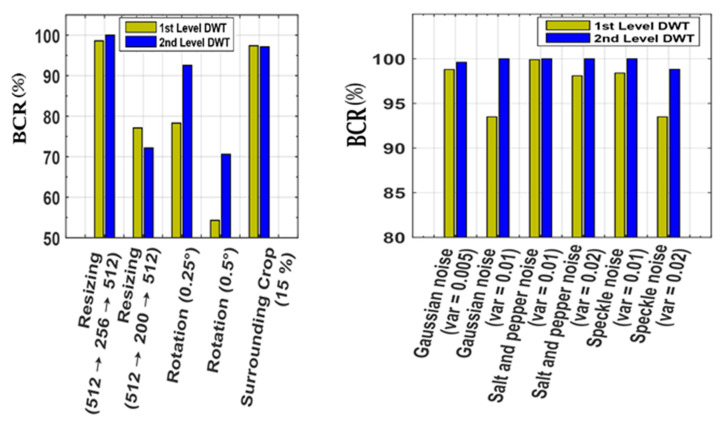
*BCR* (%) comparison of 1DWT and 2DWT for different attacks.

**Table 1 sensors-21-05540-t001:** Time complexity for an image with the size *n* = *m* × *m* image.

Step	Main Operation	Embedding Process	Extraction Process
1	DWT	*O*(*n*)	*O*(*n*)
2	Zig-zag	*O*(*n*)	*O*(*n*)
3	Split vector	*O*(*n*)	*O*(*n*)
4	DCT	*O*(*n*)	*O*(*n*)
5	Watermark embedding/extraction	*O*(*z)*	*O*(*z*)
6	Inverse DCT & Zig-zag	*O*(*n*)	-
7	Inverse DWT	*O*(*n*)	-
Total		*O*(*n*) + *O*(*z*)	*O*(*n*) + *O*(*z*)

**Table 2 sensors-21-05540-t002:** Execution time for watermark embedding and extraction processes.

Image	Image Size	Embedding Time (s)	Extraction Time (s)
Girl	256 × 256	0.1716	0.1223
Peeper	512 × 512	0.2265	0.1448
Jet-plane	512 × 512	0.2267	0.1488
Baboon	512 × 512	0.2313	0.1479
Sailboat	512 × 512	0.2251	0.1471
Boat	512 × 512	0.2311	0.1471
Pirate	1024 × 1024	0.4455	0.2457
Average		0.2511	0.1577

**Table 3 sensors-21-05540-t003:** *PSNR* (dB), *SSIM*, and *BCR* (%) measurements using different values of gain factor *α* using a 256-bit random sequence.

	*α* = 0.1			*α* = 0.2			*α* = 0.3		
Image	*PSNR*	*SSIM*	*BCR*	*PSNR*	*SSIM*	*BCR*	*PSNR*	*SSIM*	*BCR*
Peeper	46.8532	0.9903	100	41.0763	0.9637	100	37.5991	0.9276	100
Jet-plane	44.9943	0.9877	100	41.2936	0.9603	100	37.4736	0.9153	100
Baboon	45.6986	0.9960	100	40.0648	0.9834	100	37.3289	0.9712	100
Girl	40.6585	0.9583	100	35.2580	0.8834	100	31.4715	0.7825	100
Sailboat	46.7466	0.9917	100	40.9721	0.9723	100	37.3362	0.9418	100
Boat	46.5594	0.9921	100	41.1118	0.9731	100	37.5301	0.9457	100
Pirate	52.7633	0.9978	100	47.1427	0.9921	100	43.5626	0.9829	100

**Table 4 sensors-21-05540-t004:** Robustness against JPEG compression attacks in terms of *BCR* (%).

Image	Q 10	Q 20	Q 30	Q 40	Q 50	Q 60	Q 70
Girl	77.451	95.686	100	100	100	100	100
Peeper	89.124	85.458	99.986	100	100	100	100
Jet-plane	84.256	87.213	100	100	100	100	100
Bridge	90.945	100	100	100	100	100	100
Baboon	87.2541	99.845	100	100	100	100	100
Pirate	88.4575	98.356	100	100	100	100	100

**Table 5 sensors-21-05540-t005:** Robustness against various types of noise attacks in terms of *BCR* (%).

Image	Gaussian Noise (var = 0.0005)	Gaussian Noise (var = 0.001)	Speckle Noise (var = 0.01)	Speckle Noise (var = 0.02)	Salt and Pepper Noise (var = 0.01)	Salt and Pepper Noise (var = 0.02)
Baboon	99.6078	100	100	100	100	100
Jet-plane	99.5078	99.6142	100	98.82	100	100
Bridge	99.2157	100	100	100	100	100
Pirate	99.8876	100	100	100	100	100
Peppers	100	99.6125	100	100	100	100

**Table 6 sensors-21-05540-t006:** Robustness against filtering and bit plane attacks in terms of *BCR* (%).

Image	Bit-Plane Removal (5 bits)	Bit-Plane Removal (6 bits)	Gamma Correction (0.5)	Gamma Correction (1.5)	Histogram Equalization	Laplacian Sharpening
Baboon	99.5464	97.1247	100	100	100	100
Jet-plane	99.2154	95.2451	100	100	100	100
Bridge	100	93.1212	100	100	100	100
Pirate	100	96.6647	100	100	100	100
Peppers	99.987	100	100	100	100	100
Boat	98.451	99.4545	100	100	100	100

**Table 7 sensors-21-05540-t007:** Robustness against geometric attacks in terms of *BCR* (%).

Image	Rotation (0.25°)	Rotation (0.5°)	Resizing (512 → 256 → 512)	Resizing (512 → 200 → 512)	Surrounding Crop (15%)
Baboon	92.549	70.588	100	72.1569	97.1201
Jet-plane	99.607	80.784	100	60.3922	97.0214
Bridge	97.647	75.686	100	69.3725	99.1041
Peppers	100	83.529	100	71.3725	98.5102
Boat	97.254	77.647	100	72.1569	98.8564
Tree	100	90.5882	100	65.1245	96.6548

**Table 8 sensors-21-05540-t008:** Robustness against different filtering operations (*SSIM* and *BCR*%).

	*SSIM*					*BCR*				
Images	Gaussian Filter (5 × 5) var = 1.5	Gaussian Filter (5 × 5) var = 1	Median Filter (3 × 3)	Wiener Filter (3 × 3)	Average Filter (3 × 3)	Gaussian Filter (5 × 5) var = 1.5	Gaussian Filter (5 × 5) var = 1	Median Filter (3 × 3)	Wiener Filter (3 × 3)	Average Filter (3 × 3)
Girl	0.8848	0.8843	0.8785	0.8809	0.8769	100	100	100	100	100
Baboon	0.988	0.9861	0.9873	0.9875	0.9875	100	100	100	100	100
Jet	0.9645	0.957	0.9589	0.9567	0.9574	100	100	100	100	100
Bridge	0.9853	0.9828	0.986	0.984	0.9842	100	100	100	100	100
Pirate	0.9912	0.9906	0.9911	0.9898	0.9906	100	100	100	100	100
Pepper	0.9658	0.966	0.9673	0.9681	0.9668	100	100	100	100	100
Boat	0.9698	0.9666	0.9683	0.9711	0.9695	99.216	100	100	100	100
Tree	0.8965	0.9666	0.9034	0.8964	0.8952	99.216	99.2157	100	100	99.608

**Table 9 sensors-21-05540-t009:** *BCR (%)* values of 1DWT and 2DWT for different filtering attacks.

Attack	1DWT	2DWT	Attack	1DWT	2DWT	Attack	1DWT	2DWT
Average filter [5 × 5]	51.0763	99.6086	Wiener filter [5 × 5]	79.4521	100	Median Filter [5 × 5]	57.1429	97.456
Average filter [7 × 7]	17.2211	89.0196	Wiener filter [6 × 6]	73.1898	98.4344	Median Filter [6 × 6]	38.7476	91.9765
Average filter [6 × 6]	31.5068	90.6067	Wiener filter [7 × 7]	70.6458	95.3033	Median Filter [7 × 7]	24.0705	82.5832

**Table 10 sensors-21-05540-t010:** Comparison of *BCR* (%) values of proposed with existing methods.

Attack	Zermi et al., 2021 [58]	Fares et al., 2021 [31]	Ferda et al., 2018 [54]	Feng et al., 2010 [55]	Lin et al., 2008 [57]	Jiang et al., 2013 [56]	Mardolkar et al., 2016 [59]	Zhang et al., 2019 [60]	Fan et al., 2020 [42]	Proposed Method
JPEG compression 20			49.80	69.80	94.50	50.88	59.37	99.83	99.87	99.85
JPEG compression 30		98	71.97	78.40	98.40				99.87	100
JPEG compression 40	92		83.50	89.20	100	76.95	69.85	99.97	99.90	100
Gamma Correction	97			99.20	76.10					100
Gaussian noise (0.01)			99.06	87.50	79.50	70.41	68.26		99.65	100
Salt and Pepper (0.01)	99		99.17			79.98	81.74	99.84		100
Salt and Pepper (0.02)		91		93.30	93.30	69.04	67.19	99.69		100
Median Filter (3 × 3)			99.01	97.10	99.20				99.87	100

**Table 11 sensors-21-05540-t011:** Comparison of average *PSNR* (dB) values of the proposed with existing schemes.

Feng et al. [55]	Lang et el. [61]	Lin et al. [16]	Guo et al. [62]	Abdulrahman et al. [45])	Moosazadeh et al. [63]	Anand et al. [64]	Hu et al. [65]	Proposed Method
37.72	40.021	36.425	40.321	37.0035	40.76	44.19	40.48	46.85

## Data Availability

Databases used in this study are available online.

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
