# Peer review of "Encryption Based Image Watermarking Algorithm in 2DWT-DCT Domains"

_sensors, 2021, doi:10.3390/s21165540_

Round 1
Reviewer 1 Report
Authors of this paper introduce an encryption based image watermarking scheme employing a combination of Second-level Discrete Wavelet Transform (2DWT) and Discrete Cosine Transform (DCT) with an auto extraction feature. The performance of the proposed method is measured by well-known objective metrics, namely PSNR and SSIM, and partly compared with state-of-the-art (SOTA) solutions. Unfortunately, the topic of this manuscript does not fall into the wider scope of the Sensors journal.
The article, in general, is united. However, it has several shortcomings. Next, it is not clear that how it connected with the scope of journal Sensors. Unfortunately, I cannot recommend the article in the presented form for publication. Next, I recommend for the authors to carefully rework the article and try resubmission into other more appropriate journal, for instance Applied Science.
Comments:
- Article – sometimes, white space between the words and links on references is missing (e.g. “from embedded data[22]”). Please, check the whole article carefully once again!
- Introduction – the main contributions of the article are presented on general level. I recommend for the authors to highlight them (the using of items is recommended), mainly from the viewpoint of the state-of-the-art (SOTA) presented in Section 2.
- Section 2 – the SOTA is well elaborated and there are considered many works. However, there are not presented any conclusion from the provided survey. What is the main contribution of this work compared to the presented ones?
- Section 3 – the formal presentation of Figure 2 should be improved – the gray rectangular around the figure should be omitted. Next, legends should be placed inside the graph.
- Section 3 – Fig. 2 – it is not clear that how was the results obtained (simulation model, the used SW/HW configuration, time consumption – simulations, etc…). Next, the meaning of sub-bands in Fig. 1 is not clearly explained (mainly for non-expert readers).
- Section 3 – Fig. 4 “ZigZag” versus text “zigzag” – it should the same (Zig-Zag)
- Section 3 – the proposed watermark embedding process should be presented and discussed in details. It is introduced on general level.
- Section 3 – for reproducible research, I would suggest the authors make the code of their proposed method publicly available if possible
- Section 4 – variables in the text – from the equations (e.g. in (7)) – should be written by italics
- Section 4 – Figure 7 and Table 1 – from the viewpoint of PSNR, the information is redundant (the same information is presented in two different form). Next, there is a typo. The PSNR value picture Baboon at embedding coefficient=0.2 is around 35 dB in Fig. 1, but this value in Table 1 is around 40 dB. This is only one typo – please, check the whole Table 1 once again! Thanks!
- Section 5 – Tables related to BCR – it should be indicated that BCR is in units %
- Section 5 – Figure 11 – why the values of SSIM are lower for images Jet-plane and Peppers?
- Section 5 – Figure 13 should be replaced by a table (Note: units of PSNR in Fig. 13 are missing)
- Section 5 – it seems that authors compare the performance of their proposed method in terms of PSNR with other methods than in Table 8. Why? Comment “…however, the PSNR (imperceptibility) and BCR values were not satisfactory compared to the proposed method” is valid only for [55] or is not?
- Article – the formal presentation of the figures should be improved (some figures are blurred)
Reviewer 2 Report
-Some more information needed on : “AC coefficient”, p.6, line 247
-Define the : “optical level watermarking” p.6 ,line 250
-Check for typos: “for the and 0.3” p.10, line:352
-Check the equation number: “.. a direct pixel coefficient (equation 3)” p.13, line 433. It appear that equation (3) is for the two parallel vectors p.8, line 284
Reviewer 3 Report
This paper presents 'Encryption based Image Watermarking Algorithm in 2
2DWT-DCT Domains'. Overall, the presented idea is novel and interesting. A new watermarking scheme using 2DWT and DCT is proposed. Results are also compared with a number of another state of the art schemes. I have few questions:
- The proposed scheme is tested on grey-scale images. Does it work for colour images? If it does not work, how we can change it for colour images?
- Host and watermark images size should be always the same? Is there any scheme in which host image size is less?
- Though comparison has been done. Why this scheme is not compared with any of the schemes presented in 2021?
- The conclusion part should be rewritten. For example, please provides a sense of closure.
Round 2
Reviewer 1 Report
Many thanks for your response on my comments! The article has been improved.
After the check of its revised version, I have the following comments:
- Article – I respect your argumentation. On the other hand, please, explain how your work is related with “Sensors”!
- Introduction – it should be “The main contributions of this work can be summarized as follows:”
- Introduction – “the main contributions of the work” should be written in “passive form” not in the “past one”.
- Section 2 – it is written: “It has been observed that the majority of existing methods are incapable of satisfactory recovery of the watermark [31, 42]…” – is it valid only for these two works?
- Section 2 – Fig. 2 – some questions are not answered: information about the used SW/HW configuration, time consumption is missing
- Section 4 – authors make available the code of their model (thanks for this). I have checked the script and seems to be not complicated or complex (only one two m-files). I recommend for the authors to extend the article with discussion about the complexity of other available solutions in the literature. What are the main advantages/disadvantages of your model?
- Section 4 – is Table 1 (PSNR) reflects the results presented in Fig. 7? If yes, then it is redundant information. The PSNR results obtained for “gain factor” and “embedding capacity” are very similar. Please, explain it!
- Comments 12 and 14 – please, extend the article with your responses on these comments!
Author Response
Please see the attachment.

This manuscript is a resubmission of an earlier submission. The following is a list of the peer review reports and author responses from that submission.
Round 1
Reviewer 1 Report
This paper is worth for acceptance, novelty of the idea seems interesting and small changes need to be incorporated in order to enhance.
All the acronyms should be defined and explained first before using them such that they become evident for the readers.
The paper needs to be restructured in order to be precise.The Introduction and related work parts give valuable information for the readers as well as researchers. In addition recent papers should be added in the part of related work.
As it is real time application oriented, authors should care over the outcome of the proposed framework by meeting the future requirements too.
Representation of figures needs to be improved.
Grammatical errors should be validated. Most of the typos and incorrect grammars have been corrected, but it is still necessary to subject the paper to proofreading.
It would be good if similar domains, such as adversarial examples, would be reflected in future research or related work.
[1] Kwon, Hyun, Hyunsoo Yoon, and Ki-Woong Park. "Multi-targeted backdoor: Indentifying backdoor attack for multiple deep neural networks." IEICE Transactions on Information and Systems 103.4 (2020): 883-887.
Reviewer 2 Report
In this paper, authors proposed an image watermarking scheme based on 2DWT and DCT. The watermarking extraction is blind and the watermark is robust for several image processing attacks, such as JPEG compression, filtering, adding noise, cropping, sharpening, and bit-plane removal. Some experiments are carried out the verify the feasibility and performance. However, the description of this paper is a little miscellaneous and needs to be further concise. There are several problems that need further explanation.
- There are four sub-bands in each level of wavelet transform.Why choose LL2 to embed watermark?What is the theory basis?
- What is DCT's role in this watermark scheme? What is watermark data? Can the effect of watermark reconstruction be displayed intuitively?
- The flow chartsneeds to be further clarified.
- References need to be cited in order.
Reviewer 3 Report
This paper presents 'Blind image watermarking algorithm based on differential encryption in 2DWT-DCT domains'. The presented scheme is based on 2DWT and DCT with an auto extraction feature. Several image attacks, e.g., JPEG compression, filtering, noise addition, cropping, sharpening, and bit-plane are also evaluated and robustness has been proved. However, I have a few questions and suggestions.
- Please discuss numeric results in the abstract.
- Please provide Ref for 'The success of watermarking
and de-watermarking processes depends on successful retrieval of hidden data'. - In the related work section, please discuss the latest work (2021). In literature, DCT based encryption are also available, please discuss in Lit review https://link.springer.com/article/10.1007/s00521-016-2405-6
- Please provide Ref for Eq 1.
- Please define the x-axis and y-axis in Fig 1 (Histogram).
- Please adjust Fig. 4.
- Discuss some future directions.
Round 2
Reviewer 2 Report
I don't recommend this paper to be published in Sensors.
Sub band LL2 is not suitable for embedding watermark because it contains important information of the host image and data being hidden in LL2 will introduce the worst imperceptibility.
The role of DCT is not very appropriate. DCT and DWT are just simple splicing.
The references are still confused. Reference 10 is not introduced in order, and references 18, 29 and 62 contain two references.
Reviewer 3 Report
The Paper is revised based on my comments. I recommend this paper for publication
Author Response
Dear Reviewer,
Thank you for your valuable time in reviewing our paper.
Your comments enabled us to greatly improve our manuscript. We have proofread the manuscript to improve the language.
Regards,
Md Saiful Islam